# Challenges to generating political prioritization for adolescent sexual and reproductive health in Kenya: A qualitative study

Maricianah Atieno Onono[1,2]*, Claire D. Brindis[2,3,4], Justin S. White[2,3], Eric Goosby[2], Dan Odhiambo Okoro[5], Elizabeth Anne Bukusi[1], George W. Rutherford[2]

1 Center for Microbiology Research, Kenya Medical Research Institute, Kisumu, Kenya, 2 Institute of Global Health Sciences, University of California, San Francisco, California, United States of America, 3 Philip R. Lee Institute for Health Policy Studies, University of California, San Francisco, California, United States of America, 4 Adolescent and Young Adult Health National Resource Center, San Francisco, California, United States of America, 5 Nairobi country office UNFPA, Nairobi, Kenya

* maricianah@gmail.com

**Data Availability Statement:** Data are available from Dryad: (https://doi.org/10.7272/Q6F47MB2).

**Funding:** The authors received no specific funding for this work.

## Abstract

### Background

Despite the high burden of adverse adolescent sexual and reproductive health (SRH) outcomes, it has remained a low political priority in Kenya. We examined factors that have shaped the lack of current political prioritization of adolescent SRH service provision.

### Methods

We used the Shiffman and Smith policy framework consisting of four categories—actor power, ideas, political contexts, and issue characteristics—to analyse factors that have shaped political prioritization of adolescent SRH. We undertook semi-structured interviews with 14 members of adolescent SRH networks between February and April 2019 at the national level and conducted thematic analysis of the interviews.

### Findings

Several factors hinder the attainment of political priority for adolescent SRH in Kenya. On actor power, the adolescent SRH community was diverse and united in adoption of international norms and policies, but lacked policy entrepreneurs to provide strong leadership, and policy windows were often missed. Regarding ideas, community members lacked consensus on a cohesive public positioning of the problem. On issue characteristics, the perception of adolescents as lacking political power made politicians reluctant to act on the existing data on the severity of adolescent SRH. There was also a lack of consensus on the nature of interventions to be implemented. Pertaining to political contexts, sectoral funding by donors and government treasury brought about tension within the different government ministries resulting in siloed approaches, lack of coordination and overall inefficiency. However, the SRH community has several strengths that augur well for future political support. These

**Competing interests:** The authors have declared that no competing interests exist.

include the diverse multi-sectoral background of its members, commitment to improving adolescent SRH, and the potential to link with other health priorities such as maternal health and HIV/AIDS.

## Conclusion

In order to increase political attention to adolescent SRH in Kenya, there is an urgent need for policy actors to: 1) create a more cohesive community of advocates across sectors, 2) develop a clearer public positioning of adolescent SRH, 3) agree on a set of precise approaches that will resonate with the political system, and 4) identify and nurture policy entrepreneurs to facilitate the coupling of adolescent SRH with potential solutions when windows of opportunity arise.

## Introduction

There is an increased focus on adolescent sexual reproductive health (SRH) in the global health agenda [1, 2]. Several global calls including the Every Woman Every Child Global Strategy for Women's, Children's, and Adolescents' Health (2016–2030) [3] and the 2030 Agenda for Sustainable Development [4] emphasize the need to focus on adolescents. Many African states recognize the pivotal role of addressing adolescent SRH not just in achieving the Sustainable Development Goals (SDGs) in 2030, but also in reaping the demographic dividend [5]. Unfortunately, these global and regional norms and instruments are often overlooked and there is often inadequate policy orientation and political prioritization to meet adolescent SRH at the individual country level in sub-Saharan Africa. Political priority is present when: 1) national political leaders publicly and privately express continued concern for an issue, 2) the government legislates policies that offer widely accepted strategies to address the problem, and 3) the government apportions and releases public budgets proportionate with the problem's severity [6].

Priority setting for health interventions is one of the most challenging and complex issues faced by health policy decision-makers all over the world [7, 8]. Priority setting is defined as the process by which decisions are made on how health care resources should be allocated among competing programs or individuals [9]. A recent systematic review [10] found that regardless of the context, priority setting is often value-laden and political [11–14] and requires credible evidence, strong and legitimate institutions, and fair processes [15–17]. In many instances, particularly in developing countries, the priority setting process is often "messy", "ad hoc," and happens by chance [18]. In resource-limited settings such as sub-Saharan Africa, priority setting on domestic issues is often further complicated by: 1) financial constraints that create an increasing gap between available resources and demand for health services; 2) lack of sufficient and dependable data and information systems to substantiate investments in health care compared to alternative investments such as infrastructure; 3) multiple international players who provide financial and technical assistance but also have their priorities; and 4) implementation obstacles, such as political instability, conflicting political priorities, social inequalities, and inadequately developed government institutions and civil societies [7, 19, 20].

While the importance of priority setting in public health is not in question, there is a dearth of qualitative inquiry on how it is operationalized within the context of adolescent SRH and in sub-Saharan Africa. This paper qualitatively examines which factors have facilitated or

hindered political prioritization of adolescent SRH in Kenya. The conceptual model that guides this policy analysis is drawn primarily from the Shiffman and Smith framework [6], which consists of four categories: the power of actors involved, the ideas they use to portray the issue, the nature of the political contexts in which they operate, and the characteristics of the issue itself [6].

## Materials and methods

### Description of study design

We employed an interpretive focused ethnographic approach [21–23]. Ethnography seeks to develop an in-depth understanding of how people or societies make sense of their lived experience within their sociocultural environments [24]. Ethnographic methodology was well suited to this study because it allows for exploration and understanding of both the process and outcome of adolescent SRH policy making through complete observation, reconstruction, and analysis in a real-world context. Our reporting is in line with the consolidated criteria for reporting qualitative research guidelines (see S1 File) [25].

**Study setting.** The study took place in Kenya. Kenya has shown leadership in the area of adolescent SRH by adopting favorable international and regional policies and legal frameworks that promote adolescent SRH. At the global level these include the 1994 United Nations at the International Conference on Population and Development (ICPD) [26], the 2002 United Nations General Assembly Special Session on Children [27]; the Committee of the Convention on the Rights of the Child: Comment no. 4 of 2003[28]; the Convention on the Elimination of All Forms of Discrimination Against Women (CEDAW) [29] and the international Sustainable Development Goals at a global level [30]. At the regional level, in Africa, Kenya has adopted the Maputo Protocol [31] and the Common Africa Position (CAP) on the Post-2015 development agenda [32]. Locally the 2010 Constitution of Kenya, the National Youth Policy (2007) and the National Adolescent Sexual and Reproductive Health Policy (2015) all emphasize this commitment. However, adolescents in this country continue to be burdened by negative SRH outcomes. At the national level, 103 out of every 1000 births are to 15-to-19-year-old girls, which represents 37% of the national total fertility rate[33]. Data from the National AIDS Control Council show that adolescents between the ages of 15–24 years have 46% of all new infections in Kenya and represent about 17.7% of persons living with HIV and 11% of all HIV-related deaths in the country[34].

**Theoretical underpinning: Shiffman and Smith framework.** Although the Shiffman and Smith framework was originally focused on priority setting at the global health level, it has grown to be applicable in explaining the political prioritization processes in numerous national and subnational settings. In particular, this model has been used in mapping priority setting processes for health in low-resource settings across Asia, Latin America, and sub-Saharan Africa, which demonstrates its transferability to the study of health policy in resource-constrained settings such as Kenya [35, 36]. Table 1 outlines in details the main components of the framework as outlined by Shiffman and Smith[6].

**Recruitment.** We used purposive sampling to identify participants. Eligibility criteria included state and non-state policy actors in Kenya who are involved in the adolescent SRH policymaking process. State actors that were targeted included senior government officials from the ministries of health, youth and gender affairs, devolution and planning, and education. A list of potential participants was developed and prioritized according to the following criteria: job position that was previously or currently held, expected expertise and knowledge that they possess regarding SRH, and names that were repeatedly identified as being critical people to interview. We excluded officials from the sub-national governments since within

**Table 1. Shiffman and Smith framework.**

|  | Description | Factors shaping political priority |
|---|---|---|
| Actor power | The strength of the individuals and organizations concerned with the issue | 1. Policy community cohesion<br>2. Leadership<br>3. Guiding institutions<br>4. Civil society mobilization |
| Ideas | The ways in which those involved with the issue understand and portray it | 1. Internal frame<br>2. External frame |
| Political contexts | The environments in which actors operate | 1. Policy windows<br>2. Global governance structure |
| Issue characteristics | Features of the problem | 1. Credible indicators:<br>2. Severity (the size of the burden relative to other problems)<br>3. Effective interventions |

This table is a summary of the original Shiffman and Smith framework[6]

Kenya's devolved health system; policymaking is a national function. The lead researcher and a representative from the Ministry of Health's Division of Reproductive and Maternal Health identified potential participants. Participants were then contacted via telephone, given a brief overview of the study, and asked if they were willing to participate. Subsequently, interviewees were also asked to suggest other potential participants who had contributed or influenced the adolescent SRH policy making processes.

Our ethnographic methodology precluded *a priori* sample size estimation; however, for planning, we estimated that we would need to conduct in-depth interviews with approximately 15–20 individuals before reaching a data saturation point. Emphasis was placed on ensuring that there were equal numbers across a range of state and non-state actors. Recruitment continued until saturation was reached.

**Data collection.** The ethnographic approach allows for utilization of a wide range of data collection and analytical methods [37]. In adopting this approach, we undertook the following activities: 1) reflective field notes, 2) primary qualitative data using semi-structured in-depth interviews (IDIs), and 3) memoranda to keep track of any emerging theoretical insights throughout the data collection process. Interviews were conducted in English, lasted approximately 90 minutes, and were digitally recorded and transcribed. The in-depth interview guide used in the study is included in S2 file and included questions on 1) the current priority for adolescent SRH in the health agenda of Kenya, 2) how adolescent SRH fits in the key health priorities for Kenya, 3) who is responsible for setting major national health policy and who holds significant influence over these decisions, 4) what sources within Kenya, if any, put pressure on policy makers to have them increase resource allocation for adolescent SRH, and 5) how adolescent SRH should be framed to political leaders in order to generate political support.

**Data management and analysis.** All interviews were conducted in a private location at the participant's discretion by a trained and experienced qualitative researcher. Interview transcripts were transcribed by a professional transcriber prior to analysis. The interview transcripts were read and reread carefully to identify emerging codes and categories. In keeping with an ethnographic approach; data collection and analysis occurred concurrently and in an iterative manner. The data were analyzed using a theory-informed thematic analytical approach [38] using Dedoose qualitative software. Transcripts were coded paragraph by paragraph by two researchers. Consistency of coding between the two researchers was established by initially coding the same transcripts and through frequent discussion between coders until

consistency was fully established. Emerging codes were clustered into themes guided by both the core concepts emerging out of the data [39] as well as literature, background reading, researchers' experience in SRH policy making, and field notes from the reflective practice and memoranda. We employed a constant comparative approach and explored the relationships between the discussion of sensitive data and contextual situation [40]. An effort was made to ensure that the emergent codes and themes remained close to both the data and relevant literature. Throughout data collection and analysis, we practiced reflexivity by continually examining our own biases as former and existing members of the national adolescent technical working group, preferences, and theoretical perspectives and how those factors played a role in our understanding and interpretation of the processes and data we were analyzing [24].

**Ethical considerations and protection of human subjects.** The research was reviewed and approved by the Scientific and Ethics Review Unit (SERU Study 3738) at the Kenya Medical Research Institute (KEMRI) and the Committee for Human Research of the University of California, San Francisco (UCSF). All participants provided written informed consent prior to the interview being conducted. The digital audio recording of the in-depth interviews were not initiated until after the informed consent process was complete, the participant had agreed to the recording, and any initial introductions that might include identifying information had been completed. Participants were not reimbursed for participating in the study.

## Results

A total of 14 participants participated in this study (see Table 2 for institutional characteristics). The interviews took place between February 2019 and April 2019. The themes were clustered around the Shiffman and Smith framework domains. Below we highlight through rich narratives, the barriers and facilitators of generating political priority for adolescent SRH. Quotes were selected because they were typical across many persons interviewed. The listing of various IDIs before a verbatim quotation correspond to respondents who had similar views to the point being made.

**Table 2. Institutional affiliations of subjects.**

| No. | Name | ID | Type of Actor |
|---|---|---|---|
| 1. | Ministry of Health: Family planning program officer | Female | Government |
| 2. | Ministry of Health: Family planning program manager | Male | Government |
| 3. | United Nations Population Fund | Male | International Development Agency |
| 4. | Population Council | Male | International NGO |
| 5. | Sexual Reproductive Health and Rights Alliance | Male | Civil Society Organization |
| 6. | Kenya Medical Training College, Nairobi | Female | Government (Ministry of Education) |
| 7. | PATH international | Female | International NGO |
| 8. | Inter Religious Council of Kenya | Male | Civil Society Organization |
| 9. | Ministry of Youth and Gender | Female | Government |
| 10. | National Council for Population and Development | Male | Government- State Corporation (Ministry of Devolution and Planning) |
| 11. | National AIDS and STI Control Program | Female | Government (Ministry of Health) |
| 12. | JHPIEGO | Female | International NGO |
| 13. | Youth Counselor | Female | Youth representative |
| 14. | National Organization of Peer Educators | Male | Civil Society Organization |

## Actor power

Actors influence the policy making process through their knowledge, experiences, beliefs and power[41]. Within Kenya, there is an extensive multi-sectoral network of actors ranging from local and national levels of government, non-governmental and civil society groups, as well as journalists, researchers and policy analysts. These actors are organized into several technical working groups and often chaired by Ministry of Health program managers. Within these technical working groups, the actors leverage their knowledge, experiences, beliefs, and power to adapt the international and regional norms and guidelines regarding adolescent SRH to Kenya (IDI_1, IDI_2, IDI_3, IDI_4, IDI_6, IDI_7, IDI_10, and IDI_12).

> *. . .. There is a working group of family planning, another technical working group for adolescent sexual and reproductive health, another national working group for prevention of mother to child transmission* [of HIV], *a national working group for nutrition, a national working group for gender. . .These national working groups are comprised of up to 20-30-member stakeholders from different organizations. Some of the members are donors—USAID and the like, which is very strategic. Others are government line ministries that have an interest in that area and then civil society itself. All of us work. That is one of the places where we are able to influence policy. They [ministry of health] bring actors in that sector to bring their joint wisdom to the table and agree on what are the key priorities and what is it that we need to do for Kenya (International NGO; IDI_7).*

However, as is often ubiquitous in the policy-making space, differential power existed. There was a perception that the domestication of international norms and guidelines for adolescent SRH was a donor-driven issue and did not reflect the actual priority of adolescent SRH. From the perspective of power theories, resources are an obvious source of dispositional power that the actors' use during their interactions with government to influence what issue deserves funding and political attention [41, 42] (IDI_10, IDI_2, IDI_3, IDI_11, IDI_8).

> *The agenda is donor driven in that it is the donor who says that I have money for this component, so they will fund the component* [that] *their governments are supporting. If the government of America thinks that sexual reproductive health for young people is a priority, then they will come and say we have a basket here to support this. So it is not a need that is identified by the Kenyan youth, but it is a need that is identified by the donor (Faith-based organization; IDI_8).*

The ability of domestic actors to influence political commitment mainly hinges on the degree of cohesion within the policy community [17, 18]. Respondents noted that despite agreement on adolescent SRH being a priority topic within the different technical working groups, different partners dictated what specific aspects of adolescent SRH were fundable. This tension resulted in fragmented, often conflicting, multi-sectoral approaches that paralyzed the execution of the very policies they championed *(IDI_5, IDI_12, IDI_4, IDI_10).*

> *. . . You know different donors and partners have different priorities, so you will get a donor who wants to support some programs, but they will support specific programs, for example, be it on women empowerment; some partners want to support areas of adolescent health, and they will tell you they want to support in this particular area, but if you go to other areas they will not support it. For example, the US is always very specific on the areas they want to support and if you don't go their way, then you lose the funding; so particular partners will support particular areas of health program priorities (State Corporation; IDI_10).*

Globally, policy communities have been more effective where they have had policy champions or entrepreneurs to push for their agenda [17]. Policy entrepreneurs do the process of connecting the problem with a policy solution and the political factors. The entrepreneurs 1) highlight indicators of the problem to dramatize it, 2) Push for one kind of problem definition over another–invite policymakers to see for themselves, and 3) present specific policies as the solution to a problem on the agenda 4) "Soften up" by writing papers, giving testimony, holding hearings or getting press coverage [43]. However, given the lack of cohesion within the adolescent networks and the contentious nature of the adolescent SRH, none of the respondents identified a policy entrepreneur of adolescent SRH.

### Ideas: Framing the problem

Frames are ideational lenses through which policy communities define problems and their potential solutions. A good frame is one that: 1) portrays the severity of the problem, 2) presents the problem as one which can be solved if attention is given, 3) demonstrates the adversity of non-intervention, and 4) is concerned with equality and the realization of human rights. Fundamentally, adolescent SRH policy community members in Kenya hold conflicting views concerning what age range comprises adolescents. Recent literature has highlighted this problem as well [44]. The United Nations has defined an adolescent as being between 10–19 years old. Invariably, the 10-year-old is still viewed as a child, while the 19-year-old as a young adult [44]. In addition, adolescents are a heterogeneous group whose needs differ by age, whether they are in school, living with parents, are married or have children of their own. Respondents highlighted that this issue had hamstrung the effectiveness of the policy community (IDI_2, IDI_5, IDI_11, IDI_3, IDI_6).

*Sometimes we have the challenges when it comes to the definition of who is a young person, who is an adolescent? That definition is bringing a lot of problems in this country where even among the stakeholders and policymakers, it is not easy for them to agree on the classification of who is an adolescent? Who is a young person? (Civil Society Organisation; IDI_5).*

Inherent in defining the adolescent as a child is that they should not be engaging in sex [45, 46]. While many acknowledged the magnitude of teenage pregnancy, early marriage, female genital mutilation, and HIV, some members felt that the issue regarding pregnancy was one of individual self-agency and not an issue that required political attention (IDI_1, IDI_2, IDI_3, IDI_5, IDI_6, IDI_8, IDI_I0, IDI_11).

*It is a tricky question. I think the first thing is that these are adolescents; people don't believe that. . .like let us now say teenage pregnancy, as an adolescent, why in the first, should you be getting pregnant? People would be thinking that you have now started investing more in life [sex] then the adolescents will think it is normal. That is why you are finding various groups do not want the issue of comprehensive sexuality education in the school because it is like we are encouraging it; it is like a normal thing. So I think that both culturally and religiously, there is that feeling that if you invest more [in comprehensive sexuality education] then, they will now know that it is their right (State Corporation; IDI_I0).*

Respondents bemoaned the fact that political leaders primarily focused and financed other health issues, such as HIV, malaria, and maternal and child mortality, which have political and emotional appeal that adolescent pregnancy does not have (IDI_1, IDI_3, IDI_4, IDI_5, IDI_6, IDI_8, IDI_11).

*Those areas* [infectious diseases] *are well resourced because of the challenge of how infectious diseases affect everyone in the community. When it comes to SRH, they will only affect that small cohort, although now, because of the realization that HIV is common and highly prevalent within this group, they are trying to do something about it. But because of the perception of it as being "your own problem", it is not seen as the problem of the whole society, you are left with your teenage pregnancy. But when it comes to infection, then everybody cares about it (International NGO; IDI_4).*

The challenge in arriving at an acceptable framing can be attributed to the multi-sectoral nature of adolescence. The adolescent in general, cuts across national, community, household, and individual boundaries. While this produces a large network of collaborators, on the downside, it generates difficulties in consensus and definitions of problems and an external position that can generate political support. Members of the adolescent SRH community expressed challenges in framing the issue in a way that did not alienate one or more stakeholder groups (IDI_1, IDI_3, IDI_4, IDI_5, IDI_7, IDI_8, IDI_9).

*When you frame it* [adolescent SRH] *in the context of population, politicians are not interested. They want numbers; they want people to have many children, which is completely contrary. The current formula for funding for counties is population-based. So it has actually worked against us. So, we are learning that may be the way to frame it—is to talk about healthy timing and spacing of pregnancy. You want to frame it in a manner that doesn't create the impression of you controlling numbers. You want to talk about unintended pregnancy so that the church doesn't have a problem with you. It is not just the politicians; the faith-based groups also have a problem with the way you frame it. So you want to frame it in non-threatening language, but you still get the message across. You want to talk about waiting to get pregnant, in Turkana, that is what they say; the groups that work there. They say that they do not talk about family planning because young people are not planning families; they definitely do not want to have children at that age, and they just want to live their lives and have fun and do all the things that young people do. Having a family is not one of the things they are planning. So the word family planning in relation to young people is a misnomer. So you can talk of contraception, you can talk of healthy timing and spacing of pregnancy, or you can talk of waiting to get pregnant (International NGO; IDI_ 7).*

The inability to advance a cohesive public positioning of a problem often translates into disagreements over which priority interventions are acceptable [6, 47]. Generally, in order to achieve political support for any policy, there must be a coupling of a well-defined problem with a proposal of a solution that is perceived as technically feasible, compatible with policy-maker's values, reasonable in cost, and appealing to the public [47, 48] (IDI_1, IDI_3, IDI_4, IDI_7, IDI_9, IDI_12).

*Generally, the community recognizes the burden or the challenge caused by some of the issues in terms of adolescent sexual reproductive health. However, some of the interventions are not generally accepted at the community level. They recognize the challenge, but when you try to introduce this, then they say, "We are against this." There is an outcry about teenage pregnancy, for example. The community will say that teenage pregnancy is high, but they will not generally accept access to information and services (comprehensive sexuality education in the school) to favour the young people (International Development Agency; IDI_3).*

## Issue characteristics

Several issue characteristics add complexity to the political prioritization of adolescent SRH. First, is social construction: how political stakeholders view a target population in terms of its ability to exercise political will through voting and generating wealth to support these efforts. Schneider and Ingram, posit that the design, selection, and implementation of a public policy aimed at addressing a social issue can be linked to the social construction of the target population of that policy[49]. In Kenya, the age one can get an identity card, get a job and also vote is 18 years. Adolescents, who are below the age of 18 are seen as dependents and not wielding any political power that can benefit politicians and public officers and as such their issues are marginalized and are often not heard or represented in agenda setting fora (IDI_3, IDI_5, IDI_6).

> . . .*The youth may not command a strong hearing politically up there. High offices are mainly the old people. The youth may not have a say because they do not have the capacity to demand for their rights. They are busy building a career. They are still in school so that time to really lobby to advocate for their rights is not there and the person with the power to make decisions are the older people (Ministry of Education; IDI_6).*

Indicators and data play an essential role in determining priorities [6]. Until recently, sex and age disaggregated program data were often not available in national and sub-national information systems for the adolescent cohort [50]. One respondent noted that the challenge with getting adolescent-specific data was because adolescent SRH outcomes could fit into many different and sometimes concurrent categories.

> *Adolescents are crosscutting. You find adolescents who are living with HIV, you find adolescents who are pregnant, and you find adolescents who are married. You find them across different categories. . . it is crosscutting (International NGO; IDI_7).*

Respondents noted that data were available at both national and subnational levels. Predominant adolescent SRH indicators of interest included: 1) HIV incidence and prevalence, 2) maternal mortality, 3) condom use, and 4) education attainment. However there were three main issues: 1) data collected routinely through the District Health Information System were of questionable quality, 2) there was a lack of capacity or willingness to use data for decision making, and, 3) the incidence and prevalence of various adolescent SRH outcomes were not perceived to be severe enough (IDI_3, IDI_4, IDI_6, IDI_7, IDI_8, IDI_12, IDI_14).

> *I think the data is available, but the extent to which we actually analyse the data and use it for decision making; I don't think we have mastered that skill yet as a country because data is entered within computer systems; it might not be accurate as well because we have a limited capacity in the people who handle that data and a lot more needs to be done to increase supportive supervision. But, even when we have that data, we don't use it to decide on the priority needs for the areas (International NGO; IDI_12).*

With regards to interventions, nearly all respondents mentioned that youth-friendly services were the solution, and, indeed, a national guideline on how to provide youth-friendly services was being developed. The Ministry of Youth noted that it had set up youth-friendly centers. However, respondents noted that youth were not involved in the design, that no local evidence had been considered, that the intervention had not been optimized for adolescents,

and that programs needed to be designed with users in mind (IDI_1, IDI_4, IDI_6, IDI_13, IDI_14).

> *If I were a pregnant teenager, I would probably queue in the antenatal clinic with other mothers. I wouldn't go to the youth-friendly centre where they will see I am a mother and so forth. So how do we take care of this service model for various cohorts or various needs? I think that is where the challenge lies. And the reason for failure to optimize the services for young people is actually because resources are not there. People have not been able to invest much more in that. Two, we have jumped into the bandwagon of the youth-friendly services and run with it without understanding other ways we can improve on it and make it work better. . . . I guess what I am trying to suggest is that there are ways we can improve the service delivery, but it is not cut and paste (International NGO; IDI_4).*

> *When you interview the young people, they say that they want their own youth-friendly services. Currently, the youth-friendly services are only at 10%. That is what they prefer, but again, when you do further research, some of them want the services integrated (Ministry of Health; IDI_1).*

## Political contexts

The political environment in which the adolescent SRH advocates operate was not conducive to sustained prioritization of adolescent SRH. The 5-year electoral cycle meant that the political environment was continually changing and adolescent SRH kept falling in and out of favor depending on the incumbent's political party. Most politicians were guided by their own cultural or religious beliefs and the desire to remain in power and thus avoided the controversies clouding adolescent SRH (IDI_5, IDI_6, IDI_7, IDI_8, IDI_9).

> *. . .Another problem that we have as a country is whereby I'm Governor Rose; I would say this is the direction we are taking as a country; this is our CIDP* [County Integrated Development Plan], *and we've agreed this is the direction we are taking. Governor Florence comes in and feels like those projections you've made and all that are Rose's and now we are going to use mine, so there is no continuity, there is no buying of what had been initially planned as much as the community had adopted it, and maybe, there was even community participation, but now you have to have fresh community participation [engagement] forums (Ministry of Youth and Gender; IDI_9).*

Partly as a result of this 5-yearly electoral cycle there were very few policy windows that opened in which policy prioritization for adolescent SRH could occur. A recent surge in pregnant teenagers sitting for their primary school examinations was a potential policy window but, in the absence of policy entrepreneurs and data, the opportunity was missed (IDI_2, IDI_4, IDI_9, IDI_11, IDI_12).

> *. . .For instance, it was just the other day, we were talking about pregnancies; alarmed that so many girls are giving birth during the [National] exams and all that. . .In November/December last year, everyone was talking about adolescent pregnancies, and we would even ask who made the girls pregnant; some would say the boda-boda* [motorycycle taxi drivers] *people are responsible, some would say they are the older men, some would say they are the teachers and term it as transactional sex. But from there, what happened? Nothing. We are waiting for another November/December, which is just less than six months away, to start again*

*crying. . .That girl who gave birth at that time again she will be either pregnant or is already pregnant. She will get pregnant this April (Ministry of Youth; IDI_9).*

Despite numerous guidelines and published road maps, there was no political commitment or reliable mechanism to earmark funds for adolescent SRH and to account for it (IDI_3, IDI_5, IDI_6, IDI_7, IDI_10).

*Resource allocation is hard. From the programs, we will collect data through the DHIS* [District Health Information System] *even through the facilities. Then, it goes to the headquarters' Ministry of Health but for it to be funded through the treasury. The money* [from treasury] *will not come* [be allocated] *because malaria was high in Kilifi or Homa Bay; that now you will get more funding because of that, no. They do not use data so that they can give finances. They just allocate, general allocation for the roads, for the schools, for the health sector, for agriculture; it is all lumped together (Ministry of Health; IDI_1).*

Moreover, even though adolescent SRH had been incorporated into nearly all line ministries including labour, agriculture and education, some ministries lacked the know-how to implement or enforce some of the recommendations unless they were clearly aligned with the primary scope of the particular minister's office (IDI_1, IDI_3, IDI_4, IDI_7, IDI_9, IDI_12).

*I am in the youth sector. Were it not for my own interest in matters of health, I would not know so much. For example, the Ministry of Education, Ministry of Agriculture both have ways on how they can integrate* [adolescent SRH] *into their programs that are targeting youth. Then, let us look at the gender sector—they have people, but what is their level of understanding?. . .How do they* [Ministry of Health] *build the capacity of other sectors to understand, especially those who have a direct link or correlation with adolescent SRH and build their capacity to better understand matters of adolescent SRH and so they work together? You can go to the agriculture ministry and start telling them to integrate adolescent SRH only for them to ask you what it means. How do you mix sex issues with agriculture? There are some people who are not interested in all that—they only know of animal husbandry or plant husbandry, if there is anything like that. The other things, they have no interest about, and yet you have to integrate them and indirectly these are human resources, aren't they? (International NGO; IDI_12).*

Ultimately, the lack of cohesion among the network of adolescent stakeholders, their differential powers coupled with the absence of a clear public framing of the problem, lack of nuanced and credible adolescent metrics and the lack of policy (individual and institutional) entrepreneurs, manifested in having multiple editions and revisions of guidelines and policies on paper, but for which there was no tangible implementation (IDI_2, IDI_I7, IDI_11, IDI_12).

*Kenya is one country that has guidelines and policies for everything: adolescent health, family planning, HIV/AIDS, prevention of mother to child transmission. It is not the lack of documentation, meaning that we have sat and thought about it more than once. In many cases, when you look at the documents in the ministry of health libraries, you will find that it is onto the third version of the document. We are onto our second adolescent sexual and reproductive health guidelines and the second version of adolescent sexual and reproductive health policy. So it means that people have thought about it. Even when you look at vision 2030, when you*

*look at the government pillars, health is one of them. . . So, I do not think it is the lack of people talking, thinking, planning, and documenting (International NGO; IDI_7).*

## Discussion

An analysis of actor power, ideas and framing, issue characteristics, and political contexts reveals that the level of political priority for adolescent SRH in Kenya remains low. The adolescent SRH actors use two main approaches to influence the national political systems: promotion of norms and inducements using financial and technical assistance [6, 51]. This collective action has resulted in the integration of adolescent SRH into national policy documents and guidelines across different sectors, such as education, youth, health, and agriculture. However, the presence of normative guidance in the form of national policy documents and guidelines has not always promoted political priority nor deliberate action that advances a shared agenda [52]. Within the life cycle model of how norms advance through a system to become an established priority, it is possible for some norms to be internalized and taken for granted to the extent that they are no longer discussed as an issue [53]. This appears to be the case in Kenya, in which adolescent SRH guidelines are into their second and third editions, with no notable prioritization or advancement of the proposed agenda reflected in previous editions of the guidelines.

Specific to actor power, there were many different actors from diverse sectors involved in deliberations regarding what is necessary in the field. In general, diverse, heterogeneous networks, such as those seen in the fields of tuberculosis and tobacco, are beneficial in enhancing the collective understanding of a problem, its solutions and its prioritization [54, 55]. However, this diversity can also hamper cohesion and agreement on what are the main priorities [56]. In this study, beyond the collective acknowledgment that adolescent SRH was a problem, there was no coherence in what was to be funded, supported with technical resources, or prioritized. Dominant actors supported only programs and projects that fit their agendas and vision, rather than considering the actual needs of the country. Unchecked, this imbalance in decision-making power, often leads to a vicious cycle of duplication, competition, and siloing of services, which weakens the health infrastructure [57]. This, in turn, undermines the prioritization of adolescent SRH by the public and by politicians.

There were important divisions within the policy community in framing adolescent SRH as a problem. Generating consensus on the internal and external framing of a problem and its solutions is critical in generating political support and governance [58]. Internal framing has to do with how the community of adolescent SRH policy actors defines the problem, while external framing refers to how this network portrays the problem to an external audience [6]. Existing framings centre on adolescent SRH as a health issue that needs prevention and treatment, a private issue that requires individual agency, or an economic concern that drains public resources. One challenge in arriving at a cohesive framing is whether adolescents are children or young adults. Crafting a policy requires nuance that takes into account these potential differences given that what a stakeholder might advocate for a 10-year old is not necessarily the same as for a 19 year old. At the political level, politicians, who are often risk-averse, may be hesitant to engage with controversial issues when there are other problems with safer and popular solutions. In Kenya, this controversy has resulted in adolescent issues being integrated in maternal and child health. Unfortunately, this integration makes it easy for actors to "pass the buck" to other external actors and assume that they will handle the problem [59]. Throughout the 1990's, this similar lack of clarity in framing and back-passing contributed to the neglect of newborn survival as a priority issue as it was traditionally sandwiched into

maternal and child health agendas. Ultimately newborn survival gained priority when stakeholders agreed to disentangle the newborn from the child and the mother as a distinct group and when stakeholders with interests beyond the health field started to engage with the issue. [60]

To realize the SRH wellbeing of adolescents and to protect their human rights, countries need to adopt holistic interventions that address adolescents' fully lived realities, rather than one-dimensional approaches or trickle down interventions that appear to be reactive rather than proactive, such as providing free maternal health care after girls are already pregnant. In the interviews, several actors mentioned that it was anticipated that the benefits of improving, for example, skilled attendance at birth and contraceptive access would trickle down to improve the delivery outcomes among adolescents, instead of primary prevention of the pregnancy in the first place. Adolescent SRH can learn from the maternal health networks, which emerged from near neglect in years before 2000 to a heightened transformative political priority attracting resource commitments in the early 2000s with the advent of the millennium development goals (MDGs). Policy scholars posit that maternal health, unlike other aspects of women's health, gained political priority in part because after many years of disagreement key actors finally agreed on a singular objective with a defined set of feasible solutions, i.e., to reduce maternal mortality by three quarters by 2015 from 1990 levels and a set of solutions that included access to emergency obstetric care and skilled attendance at birth [61]. Adolescent SRH on the other hand was only partially operationalized in the MDG by the indicator "adolescent birth rate" which tells an incomplete story [61]. Going forward, we posit that embedding adolescent-specific SRH metrics into popular international norms such as the sustainable development goals (SDGs) can trigger action and innovation towards improving adolescent SRH in Kenya. We suggest use of metrics that politicians can understand, metrics that not only measure health outcomes but also economic costs, such as cost of mortality averted or morbidity or the losses made, and how cost-effective the interventions can be.

The nature of the affected target group, coupled with the lack of credible indicators, data on its severity, and effective interventions, can significantly hamper the prioritization of an issue. To start with, a major deterrent to political attention to adolescent SRH is related to the social construction of the population. Political prioritization is more likely to emerge when the population affected wields political power (ability to vote), generates sympathy, such as children, or can mobilize itself, such as persons living with HIV and AIDS. Political prioritization may also be more likely if the problem causes high morbidity and mortality or social disruption, such as maternal health. Unfortunately, until recently in many African countries, there was a paucity of data and specific indicators on SRH behaviors of adolescents, the health and economic consequences of those behaviors, service and information needs, and effective interventions. Neonatal mortality is a good example of an issue which was neglected up to early 2000s, in part because existing vital registration systems in developing countries under-reported neonatal deaths, and it was perceived that expensive high-class interventions were necessary to ameliorate the situation. It was only when the World Health Organization released the first global estimates indicating that more than 5 million neonates had died in 1995 that priority for neonatal mortality begun to emerge [60]. Presently, there is a considerable movement to disaggregate data for adolescents by age, sex, national and sub-national levels. Kenyan actors boasted of collecting a broad range of data. This data could be critical for incentivizing actors from different sectors to form stronger collaborations and better quantification of the scope and severity of adolescent SRH. However, for political attention to be gained, there must be a coupling of the adolescent SRH problem with well-defined, feasible, cost-effective, and acceptable solutions. Although majority of the respondents talked of provision of youth friendly services as a key intervention, its implementation had not been optimized for the adolescents. Existing

reviews of what works for adolescents have frequently highlighted that effective interventions for adolescent SRH often fail or have transient effects because they are delivered ineffectively e.g. through the stand alone youth centers described by respondents, or are delivered piece-meal or with inadequate dosage[62]. In re-positioning neonatal mortality, actors had to frame it as a high-burden problem with low-technology community solutions [60]. As one respondent mentioned, within a multi-cultural and heavily "religious" context in countries such as Kenya, a simple cut and paste of interventions from other regions will not have traction with the political class that is trying to please the electorate and stay in power.

Even though policy makers may recognize the existence, severity, and repercussions of poor adolescent SRH outcomes, many policy makers are often distracted by a myriad of issues and have limited resources to deal with them alongside other conflicting political priorities. In 2015, the United Nations Population Fund (UNFPA) estimated that nearly 20% of the SRH budget in Africa was donor funded [63]. While donor funding has indeed catalyzed the recognition of adolescent SRH as a problem, the fact that it is predominantly from international organizations delegitimizes the importance of prioritizing it in Kenya [56]. Additionally, some of the external funding is sectoral in nature and hampers collaboration. The government of Kenya has integrated "youth" into nearly all its ministries. While this is in line with international norms, it has brought about tension within the different ministries resulting in non-performance or duplication of efforts and overall inefficiency. These challenges have also been seen within the early childhood development networks, which often cut across the Ministry of Education, the Ministry of Health, the Ministry of Gender, and the Ministry of Social Welfare [58]. The downside is that, although formally there is a plurality of line ministries concerned with adolescent SRH, no institutional leader, who can champion the adolescent SRH agenda across a wide variety of ministries, has emerged.

The study limitations deserve mention. One limitation of the study is that we used purposive sampling, and study participants also helped to identify other potential participants. We acknowledge that in giving the study participants this "gatekeeping role" we might have shaped the type of participants enrolled into the study, for example, by selecting potential participants who were better known. To mitigate this, we limited the role of enrolled participants in identifying only those participants who met the eligibility criteria regardless of their relationship and engagement with them. Secondly, interviews were conducted exclusively with national level stakeholders, therefore, sub-national variations in political prioritization in the devolved counties may not be adequately represented. The Shiffman and Smith framework does not address the problem of non-implementation of the policy once it has been legislated; however, it does provide the opportunity to highlight areas that can be used to raise the profile of a condition to an actionable problem.

## Conclusion

Despite a surge in interest in adolescent SRH by the global community, nations such as Kenya still fail to translate this issue into consistent political prioritization. In order for adolescent SRH to gain traction within the national political system, there is an urgent need for policy actors to use their technical and financial resources to create a more cohesive community of advocates across sectors and to develop a clear problem definition of adolescent SRH and a public positioning of the matter. This might require a compromise in the public positioning as well as range of proposed solutions to ensure that they are both palatable to the political system and thus increase tractability of adolescent SRH. There is also a need to identify and nurture individuals and national institutions that can act as policy entrepreneurs to facilitate the coupling of the problem of adolescent SRH with potential solutions when windows of opportunity

arise. In addition, non-governmental donors can increase their legitimacy as actors in the adolescent SRH space by creatively sharing their authority and control of resources with national governments.

## Supporting information

**S1 File. Consolidated criteria for reporting qualitative research (COREQ) checklist.** S1 File.
(DOC)

**S2 File. Semi-structured interview guide.** S2 File.
(DOC)

## Acknowledgments

We thank the respondents who participated in this study. We acknowledge the Director General of KEMRI.

## Author Contributions

**Conceptualization:** Maricianah Atieno Onono, Claire D. Brindis, Eric Goosby, Elizabeth Anne Bukusi, George W. Rutherford.

**Data curation:** Maricianah Atieno Onono.

**Formal analysis:** Maricianah Atieno Onono, Dan Odhiambo Okoro.

**Funding acquisition:** Maricianah Atieno Onono.

**Investigation:** Maricianah Atieno Onono, Dan Odhiambo Okoro, George W. Rutherford.

**Methodology:** Maricianah Atieno Onono, Claire D. Brindis, Justin S. White, Eric Goosby, Dan Odhiambo Okoro, Elizabeth Anne Bukusi, George W. Rutherford.

**Project administration:** Maricianah Atieno Onono.

**Resources:** Maricianah Atieno Onono, Elizabeth Anne Bukusi.

**Software:** Maricianah Atieno Onono.

**Supervision:** Claire D. Brindis, Justin S. White, Eric Goosby, Elizabeth Anne Bukusi, George W. Rutherford.

**Validation:** Maricianah Atieno Onono, Claire D. Brindis, Justin S. White, Dan Odhiambo Okoro, Elizabeth Anne Bukusi, George W. Rutherford.

**Visualization:** Maricianah Atieno Onono.

**Writing – original draft:** Maricianah Atieno Onono, Justin S. White, Dan Odhiambo Okoro.

**Writing – review & editing:** Maricianah Atieno Onono, Claire D. Brindis, Justin S. White, Eric Goosby, Elizabeth Anne Bukusi, George W. Rutherford.

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
