## [Decision Letter · Decision Letter 0]

29 Oct 2019

PONE-D-19-27067

Challenges to generating political prioritization for adolescent sexual and reproductive health in Kenya: a qualitative study

PLOS ONE

Dear Dr Onono,

Thank you for submitting your manuscript to PLOS ONE. After careful consideration, we feel that it has merit but does not fully meet PLOS ONE’s publication criteria as it currently stands. Therefore, we invite you to submit a revised version of the manuscript that addresses the points raised during the review process.

We would appreciate receiving your revised manuscript by Dec 13 2019 11:59PM. To enhance the reproducibility of your results, we recommend that if applicable you deposit your laboratory protocols in protocols.io, where a protocol can be assigned its own identifier (DOI) such that it can be cited independently in the future. For instructions see: http://journals.plos.org/plosone/s/submission-guidelines#loc-laboratory-protocols

We look forward to receiving your revised manuscript.

Kind regards,

Joshua Amo-Adjei, Ph.D

Academic Editor

PLOS ONE

Journal Requirements:

Reviewers' comments:

Reviewer's Responses to Questions

**Comments to the Author**

1. Is the manuscript technically sound, and do the data support the conclusions?

Reviewer #1: Yes

Reviewer #2: Yes

2. Has the statistical analysis been performed appropriately and rigorously? 

Reviewer #1: N/A

Reviewer #2: N/A

3. Have the authors made all data underlying the findings in their manuscript fully available?

Reviewer #1: Yes

Reviewer #2: No

4. Is the manuscript presented in an intelligible fashion and written in standard English?

Reviewer #1: Yes

Reviewer #2: Yes

5. Review Comments to the Author

Reviewer #1: The authors examine the factors responsible for lack of political prioritization of adolescent sexual and reproductive health (SRH) service provision in Kenya. The paper is generally well written and addresses an important topic. I, however, have the following comments for the authors to consider to further improve it:

1) Study setting (page 6):

a) The authors start by arguing that Kenya has shown leadership in adopting international policies and legal frameworks on adolescent SRH (first two statements). However, the examples they give in the third statement are mostly national policies except the Maputo Protocol. Given the framing of the argument, one would expect that they give examples of the international and regional policies and legal frameworks which Kenya has adopted. They should then follow this with examples of national policies that have been informed by those international instruments in a logical sequence.

b) Last statement: Change “HIV-related deaths in Kenya” at the end of the statement to “HIV-related deaths in the country” to avoid unnecessary repetition of “Kenya” in the statement. Also, provide a citation for the statement as it quotes numbers that are not from the authors’ own research.

2) Theoretical underpinning, second statement (page 7): Change “in low-resource setting countries” to “in low-resource settings” for the statement to read well.

3) Recruitment (pages 7-8):

a) First paragraph, second statement (page 7): Change “various relevancies” to “following criteria”, and delete “by” before “names” for the statement to read well.

b) First paragraph, last statement (page 8): Insert “policy” between “SRH” and “making” for the statement to read well.

4) Data collection (page 8):

a) Third statement, point #2: Change “how adolescent SRH fit in with the key” to “how adolescent SRH fits in the key” for it to read well.

b) Third statement, point #4: The argument seems to connote that pressure put on policy makers made them increase resource allocation for adolescent SRH when no evidence has been provided to that effect. Perhaps the authors should consider rephrasing it to something like, “what sources within Kenya, if any, put pressure on policy makers to have them increase resource allocation for adolescent SRH” so that it is clear that the pressure is to make policy makers increase rather than that it made them increase resources.

5) Data management and analysis (page 9):

a) Eighth statement: The authors state that emerging codes were clustered into themes guided, in part, by researchers’ values? What were these values and how did they influence coding of emerging themes?

b) Last statement: Delete “Finally” from the statement as this is not the last thing we are reading in the paper.

6) Ideas: Framing the problem (pages 13-17):

a) First paragraph, third statement (page 13): Change “comprises being an adolescent” to “comprises adolescents”.

b) Second quote (page 14): Did the participant refer to “comprehensive sexual education” or this is an artifact of transcription given that it should be “comprehensive sexuality education”? If it was the participant’s mistake, then we need to add “[sic]” at the end of the word to show that the mistake was not the authors’. The gist of the argument in the quote is also not clear. I thought most investments are to prevent unintended pregnancy among adolescents. If that is the case, how can that make pregnancy be seen as normal?

c) Third paragraph (page 14): Delete “yet” from the statement for the statement to read well.

d) Last quote (page 17): The same comment regarding “comprehensive sexual education” applies here.

7) Issue characteristics (pages 17-20):

a) Third paragraph and the subsequent quote (page 18): Delete “lastly” from the point #2 of the last statement in the paragraph since this is not the last thing we are reading in the paper. The edits aside, there is need for being specific here when referring to data of questionable quality (both in the paragraph and the subsequent quote). We have data on adolescent SRH indicators from standardized national surveys such as the Demographic and Health Surveys (DHS) and from facilities mostly through the District Health Information System (DHIS). If we talk about poor quality data, which specific sources are we referring to?

b) Fourth paragraph (page 19): Change “was in development” in the first statement to “was being developed”. That aside, what was the authors’ take on most participants’ views on youth-friendly services vis-à-vis the evidence that such interventions are not effective in improving adolescent SRH? (See, for instance, Chandra-Mouli V, Lane C, Wong S. What Does Not Work in Adolescent Sexual and Reproductive Health: A Review of Evidence on Interventions Commonly Accepted as Best Practices. Global Health: Science and Practice, 2015, 3(3):333-340).

c) Fourth quote (page 19): I did not understand what the participant meant by the statement, “And the reason is failure to optimize the services for young people, is actually because resources are not there.”

8) Political contexts (pages 20-23):

a) First quote (page 20): It is not clear what the participant meant by the phrase, “but now you have to fresh community participation forums”.

b) Fourth paragraph (page 22): Change “lacked the know how of how to implement” to “lacked the know-how to implement”.

9) Discussion (pages 23-29):

a) Third paragraph (pages 25-26): Change “10year” in the sixth statement to “10-year”, delete “issue” from the end of the ninth statement to avoid unnecessary repetition of the same word, and rephrase the last statement to read: “Ultimately, newborn survival gained priority when …”

b) Fourth paragraph (pages 26-27): Delete “a” before “trickle down” in the first statement for it to read well. Rephrase the second part of the third statement to read: “which emerged from near neglect in years before 2000 to a heightened transformative political priority attracting resource commitments …” Also, change “was able to gain” in the fourth statement to “gained”, and “they finally agreed” to “key actors finally agreed” since it was not clear what “they” here referred to.

c) Fifth paragraph (pages 27-28): Rephrase the fifth statement to read: “Unfortunately, until recently in many African countries, there was a paucity of data and specific indicators on adolescent SRH behaviors, …”

d) Last paragraph (page 29): The fifth statement needs to be appropriately formatted i.e. “In addition, interviews …” In its current format, the word “interviews” seems to start a new statement. Also, delete “lastly” from the last statement. Instead, the statement can be rephrased to something like, “The Shiffman and Smith framework also does not address …” so that it seamlessly flows from the preceding statement.

10) Conclusion (page 30):

a) First statement: Change “are still failing” to “still fail”.

b) Fourth statement: Change “are able to” to “can”.

c) Last statement: Delete “Lastly” from the statement. The statement can instead be rephrased to something like, “In addition, non-governmental donors can …”

11) Abstract (page 2):

a) Methods: Include the sample size and dates of data collection in the methods section of the abstract e.g. “We undertook semi-structured interviews with 14 members of adolescent SRH networks between February and April 2019 …” The authors should note that abstracts should be framed in such a way that they can stand alone independent of the full paper.

b) Findings, sixth statement: Change “sectorial” to “sectoral” i.e. pertaining to different sectors.

Reviewer #2: Thank you very much for inviting me to review this interesting manuscript, which focuses on prioritization of adolescent sexual and reproductive health in Kenya. It makes several findings that are quite revealing, including the lack of coordination among the civil society groups working in the space of adolescent SRH. I suggest minor revisions, which I outline in the comments to authors.

6. PLOS authors have the option to publish the peer review history of their article (what does this mean?). If published, this will include your full peer review and any attached files.

Reviewer #1: No

Reviewer #2: No

---

## [Author Response · Author response to Decision Letter 0]

1 Nov 2019

31 October 2019

Thank you for your time and careful review of our manuscript “PONE-D-19-27067 Challenges to generating political prioritization for adolescent sexual and reproductive health in Kenya: a qualitative study.” We deeply appreciate the feedback from the reviewers.

We have addressed your comments in the revised manuscript and have detailed our responses to each point below (in blue font). 

Thank you very much for your time and consideration of this revised manuscript for publication in PLOS ONE. If there is any additional information or details about the study that I can provide, please do not hesitate to contact me. We look forward to your response.

Sincerely,

Authors

Review Comments to the Author

Reviewer #1: The authors examine the factors responsible for lack of political prioritization of adolescent sexual and reproductive health (SRH) service provision in Kenya. The paper is generally well written and addresses an important topic. I, however, have the following comments for the authors to consider to further improve it:

1) Study setting (page 6):

a) The authors start by arguing that Kenya has shown leadership in adopting international policies and legal frameworks on adolescent SRH (first two statements). However, the examples they give in the third statement are mostly national policies except the Maputo Protocol. Given the framing of the argument, one would expect that they give examples of the international and regional policies and legal frameworks which Kenya has adopted. They should then follow this with examples of national policies that have been informed by those international instruments in a logical sequence.

Thank you for this direction. We have edited the section and given examples of the international and regional policies in a sequential manner. The section now reads

“At a global level these include the 1994 United Nations at the International Conference on Population and Development (ICPD)[26], the 2002 United Nations General Assembly Special Session on Children of 2002[27]; the Committee of the Convention on the Rights of the Child: Comment no. 4 of 2003[28]; the Convention on the Elimination of All Forms of Discrimination Against Women (CEDAW)[29] and the international Sustainable Development Goals at a global level[30]. At a regional level, in Africa, Kenya has adopted the Maputo Protocol [31] and the Common Africa Position (CAP) on the Post-2015 development agenda[32]. Locally the 2010 Constitution of Kenya, National Youth Policy (2007) and the National Adolescent Sexual and Reproductive Health Policy (2015) all emphasize this commitment.”

b) Last statement: Change “HIV-related deaths in Kenya” at the end of the statement to “HIV-related deaths in the country” to avoid unnecessary repetition of “Kenya” in the statement. Also, provide a citation for the statement as it quotes numbers that are not from the authors’ own research.

Thank you, we have edited to read “HIV-related deaths in the country”. We also provide a citation for this data

2) Theoretical underpinning, second statement (page 7): Change “in low-resource setting countries” to “in low-resource settings” for the statement to read well.

This has been edited to “in low-resource settings”

3) Recruitment (pages 7-8):

a) First paragraph, second statement (page 7): Change “various relevancies” to “following criteria”, and delete “by” before “names” for the statement to read well. 

This has been edited

b) First paragraph, last statement (page 8): Insert “policy” between “SRH” and “making” for the statement to read well. 

This has been edited

4) Data collection (page 8):

a) Third statement, point #2: Change “how adolescent SRH fit in with the key” to “how adolescent SRH fits in the key” for it to read well. 

This has been edited

b) Third statement, point #4: The argument seems to connote that pressure put on policy makers made them increase resource allocation for adolescent SRH when no evidence has been provided to that effect. Perhaps the authors should consider rephrasing it to something like, “what sources within Kenya, if any, put pressure on policy makers to have them increase resource allocation for adolescent SRH” so that it is clear that the pressure is to make policy makers increase rather than that it made them increase resources.

This has been edited

5) Data management and analysis (page 9):

a) Eighth statement: The authors state that emerging codes were clustered into themes guided, in part, by researchers’ values? What were these values and how did they influence coding of emerging themes?

We have clarified that its not so much values but our experiences in this field. As disclosed in the paper and in the COREQ statement, some of the researchers had been a part of the national adolescent technical working group. During coding and analysis—we purposely selected an analyst who did not have this kind of experience so as to balance out our biases. In page 10, the first paragraph, we state that “Throughout data collection and analysis, we practiced reflexivity by continually examining our own biases as former and existing members of the national adolescent technical working group, preferences, and theoretical perspectives and how those factors played a role in our understanding and interpretation of the processes and data we were analyzing [24].”

b) Last statement: Delete “Finally” from the statement as this is not the last thing we are reading in the paper.

This has been edited

6) Ideas: Framing the problem (pages 13-17):

a) First paragraph, third statement (page 13): Change “comprises being an adolescent” to “comprises adolescents”. 

This has been edited

b) Second quote (page 14): Did the participant refer to “comprehensive sexual education” or this is an artifact of transcription given that it should be “comprehensive sexuality education”? If it was the participant’s mistake, then we need to add “[sic]” at the end of the word to show that the mistake was not the authors’. The gist of the argument in the quote is also not clear. I thought most investments are to prevent unintended pregnancy among adolescents. If that is the case, how can that make pregnancy be seen as normal?

This is a transcription error and should read “comprehensive sexuality education”

The gist of this quote is that 1) people do not believe adolescents should be having sex and that any “attention/investment” in SRH (comprehensive sexuality education) for adolescents is a license to increase sexual activity among adolescents. 

c) Third paragraph (page 14): Delete “yet” from the statement for the statement to read well.

This has been edited

d) Last quote (page 17): The same comment regarding “comprehensive sexual education” applies here.

This has been edited

7) Issue characteristics (pages 17-20):

a) Third paragraph and the subsequent quote (page 18): Delete “lastly” from the point #2 of the last statement in the paragraph since this is not the last thing we are reading in the paper. The edits aside, there is need for being specific here when referring to data of questionable quality (both in the paragraph and the subsequent quote). We have data on adolescent SRH indicators from standardized national surveys such as the Demographic and Health Surveys (DHS) and from facilities mostly through the District Health Information System (DHIS). If we talk about poor quality data, which specific sources are we referring to?

This has been edited. The reference here for poor quality data was to the locally collected data in the health facilities through the DHIS. We have edited this statement to read “data collected routinely through the District Health Information System were of questionable quality”

b) Fourth paragraph (page 19): Change “was in development” in the first statement to “was being developed”. This has been edited 

That aside, what was the authors’ take on most participants’ views on youth-friendly services vis-à-vis the evidence that such interventions are not effective in improving adolescent SRH? (See, for instance, Chandra-Mouli V, Lane C, Wong S. What Does Not Work in Adolescent Sexual and Reproductive Health: A Review of Evidence on Interventions Commonly Accepted as Best Practices. Global Health: Science and Practice, 2015, 3(3):333-340). Thank you for this reference. Although the concept of implementing youth friendly services as an intervention has been shown as effective, the manner in which in which it was implemented was as stand alone youth centers and has continued to lack all the 4 elements that Chandra-Mouli et al say comprise youth friendly services. Within the discussion section we have added our thoughts as follows

“Although majority of the respondents talked of provision of youth friendly services as a key intervention, its implementation had not been optimized for the adolescents. Existing reviews of what works for adolescents have frequently highlighted that effective interventions for adolescent SRH often fail or have transient effects because they are delivered ineffectively e.g. through the stand alone youth centers described by respondents, or are delivered piecemeal or with inadequate dosage[61].” 

c) Fourth quote (page 19): I did not understand what the participant meant by the statement, “And the reason is failure to optimize the services for young people, is actually because resources are not there.” The respondent was frustrated. Implementing the youth friendly services means that 1) providers are trained to provide friendly and non-judgmental services to adolescents; 2) the health facility setting is welcoming; 3) adolescents are made aware through outreach activities about the availability of services; and 4) the larger community is supportive of provision of services to adolescents. All these require financial, infrastructural and human resource that was limited and perhaps, there was a need to “Tweak” the model for the setting as opposed to implementing it ineffectively or at inadequate dosage as Chandra-Mouli et al say in the reference provided. The next quote shows that only 10% of the services provide youth friendly services and the existing model was actually closer to youth centers than to what is envisioned in the classic youth friendly service model.

8) Political contexts (pages 20-23):

a) First quote (page 20): It is not clear what the participant meant by the phrase, “but now you have to fresh community participation forums”.

Means community engagement. I have included this in parenthesis

b) Fourth paragraph (page 22): Change “lacked the know how of how to implement” to “lacked the know-how to implement”.

This has been edited

9) Discussion (pages 23-29):

a) Third paragraph (pages 25-26): Change “10year” in the sixth statement to “10-year”, delete “issue” from the end of the ninth statement to avoid unnecessary repetition of the same word, and rephrase the last statement to read: “Ultimately, newborn survival gained priority when …” 

This has been edited

b) Fourth paragraph (pages 26-27): Delete “a” before “trickle down” in the first statement for it to read well. Rephrase the second part of the third statement to read: “which emerged from near neglect in years before 2000 to a heightened transformative political priority attracting resource commitments …” Also, change “was able to gain” in the fourth statement to “gained”, and “they finally agreed” to “key actors finally agreed” since it was not clear what “they” here referred to.

This has been edited

c) Fifth paragraph (pages 27-28): Rephrase the fifth statement to read: “Unfortunately, until recently in many African countries, there was a paucity of data and specific indicators on adolescent SRH behaviors, …”

This has been edited

d) Last paragraph (page 29): The fifth statement needs to be appropriately formatted i.e. “In addition, interviews …” In its current format, the word “interviews” seems to start a new statement. Also, delete “lastly” from the last statement. Instead, the statement can be rephrased to something like, “The Shiffman and Smith framework also does not address …” so that it seamlessly flows from the preceding statement.

This has been edited and now reads, “Secondly, interviews were conducted exclusively with national level stakeholders, therefore, sub-national variations in political prioritization in the devolved counties may not be adequately represented. The Shiffman and Smith framework…..”

10) Conclusion (page 30):

a) First statement: Change “are still failing” to “still fail”. This has been edited

b) Fourth statement: Change “are able to” to “can”. This has been edited

c) Last statement: Delete “Lastly” from the statement. The statement can instead be rephrased to something like, “In addition, non-governmental donors can …” This has been edited

11) Abstract (page 2):

a) Methods: Include the sample size and dates of data collection in the methods section of the abstract e.g. “We undertook semi-structured interviews with 14 members of adolescent SRH networks between February and April 2019 …” The authors should note that abstracts should be framed in such a way that they can stand alone independent of the full paper. This has been edited 

b) Findings, sixth statement: Change “sectorial” to “sectoral” i.e. pertaining to different sectors. This has been edited 

Reviewer #2: Thank you very much for inviting me to review this interesting manuscript, which focuses on prioritization of adolescent sexual and reproductive health in Kenya. It makes several findings that are quite revealing, including the lack of coordination among the civil society groups working in the space of adolescent SRH. I suggest minor revisions, which I outline in the comments to authors.

---

## [Decision Letter · Decision Letter 1]

14 Nov 2019

PONE-D-19-27067R1

Challenges to generating political prioritization for adolescent sexual and reproductive health in Kenya: a qualitative study

PLOS ONE

Dear Dr Onono,

Thank you for submitting your manuscript to PLOS ONE. After careful consideration, we feel that it has merit but does not fully meet PLOS ONE’s publication criteria as it currently stands. Therefore, we invite you to submit a revised version of the manuscript that addresses the points raised during the review process.

We would appreciate receiving your revised manuscript by Dec 29 2019 11:59PM. To enhance the reproducibility of your results, we recommend that if applicable you deposit your laboratory protocols in protocols.io, where a protocol can be assigned its own identifier (DOI) such that it can be cited independently in the future. For instructions see: http://journals.plos.org/plosone/s/submission-guidelines#loc-laboratory-protocols

We look forward to receiving your revised manuscript.

Kind regards,

Joshua Amo-Adjei, Ph.D

Academic Editor

PLOS ONE

Reviewers' comments:

Reviewer's Responses to Questions

**Comments to the Author**

1. If the authors have adequately addressed your comments raised in a previous round of review and you feel that this manuscript is now acceptable for publication, you may indicate that here to bypass the “Comments to the Author” section, enter your conflict of interest statement in the “Confidential to Editor” section, and submit your "Accept" recommendation.

Reviewer #1: (No Response)

Reviewer #2: (No Response)

2. Is the manuscript technically sound, and do the data support the conclusions?

Reviewer #1: Yes

Reviewer #2: Yes

3. Has the statistical analysis been performed appropriately and rigorously? 

Reviewer #1: N/A

Reviewer #2: Yes

4. Have the authors made all data underlying the findings in their manuscript fully available?

Reviewer #1: Yes

Reviewer #2: No

5. Is the manuscript presented in an intelligible fashion and written in standard English?

Reviewer #1: Yes

Reviewer #2: Yes

6. Review Comments to the Author

Reviewer #1: The authors have addressed the issues raised by reviewers. There are, however, a few editorial corrections to make. These include:

1) Study setting, second statement (page 6): Rephrase to read: “At the global level, these include …” Also, delete “of 2002” at the end of the UN General Assembly on Special Session on Children, and “at a global level” at the end of the statement since it is unnecessary repetition of the phrase used at the beginning.

2) Study setting, third statement (page 6): Rephrase to read: “At the global level …” and delete one “the” after “adopted” to avoid unnecessary repetition.

3) Study setting, fourth statement (page 6): Insert “the” before “National Youth Policy” for the statement to read well.

4) Ideas: Framing the problem, second quote (pages 14-15): Based on the authors’ response to earlier comment regarding the clarity of the quote, they need to add “[comprehensive sexuality education]” in brackets as shown after “invest more” so that it is clear that the participant was referring to investing in comprehensive sexuality education rather than in programs to prevent pregnancy among adolescents generally.

5) Issue characteristics, fourth quote (pages 19-20): Based on the authors’ response to an earlier comment regarding the quote, it should then read: “And the reason for failure ...”

6) Political contexts, first quote (pages 20-21): Based on the authors’ response to the comment on this quote, if the participant used the word “fresh” to mean conducting community forums afresh, then the authors need to insert “[sic]” after the word to show that it was the participant’s mistake (i.e. we cannot use the word “fresh” as a verb). If this was an artifact of transcription, then the authors should change “fresh” to “conduct afresh”.

7) Discussion, fourth paragraph, fourth statement (page 27): Change “from 1990 levels by 2015” to “by 2015 from 1990 levels” for the statement to read well.

Reviewer #2: These were my initial comments that were not forwarded to the authors:

"I find this to be an interesting paper and the authors have written it in very clear and easy to read language. Overall, I believe the manuscript, when published will contribute to the understanding of the policy-making and agenda-setting processes. It highlights some of the controversies around adolescence, including the definition of who is an adolescent and the age to include in that definition, the acceptability of providing comprehensive sexuality education to adolescents, the problem of acceptable language, the difficulty of civil society coordination, and the perennial problem of donor-influence in agenda-setting. There are some areas I think the work may require some revisions:

1. State and non-state actor -- It will be useful in the methods section to highlight who the state actors are. I understand that this is in the Table 1, but a high level statement to show the state actors are will be useful rather than expecting the readers to wait until they get to the table or to flip to the table to find the actors and flip back to continue reading.

2. What do the notations ID1, ID2, ID3... just before a verbatim quotation mean? First, what is the meaning of "ID"? Second, is the listing various IDs before a verbatim quotation indicative of the relevance of the interviews for a particular participant to the point being made? If it is, it will be necessary to state that that is the case. I think the referencing for the quotes can be improved by actually stating the respondent, for example, "Ministry of Youth" or some other form of referencing the quotes.

3. In a few instances, there are cases of misplaced punctuation marks, which need to be revised. For example before a verbatim quote the authors often place a fulls-stop and open a bracket to place the ID1, ID2... in without a full-stop at the end of the list.

4. It is surprising that the study could find that there are no adolescent SRH champions in Kenya. I wonder if this is simply a matter of the participants that were interviewed. It is really hard to believe that there are no champions. Could the author comment further on this?"

7. PLOS authors have the option to publish the peer review history of their article (what does this mean?). If published, this will include your full peer review and any attached files.

Reviewer #1: No

Reviewer #2: No

---

## [Author Response · Author response to Decision Letter 1]

25 Nov 2019

25 November 2019

Thank you for your time and careful review of our manuscript “PONE-D-19-27067 Challenges to generating political prioritization for adolescent sexual and reproductive health in Kenya: a qualitative study.” We deeply appreciate the feedback from the reviewers.

We have addressed your comments in the revised manuscript and have detailed our responses to each point below (in blue font). 

Thank you very much for your time and consideration of this revised manuscript for publication in PLOS ONE. If there is any additional information or details about the study that we can provide, please do not hesitate to contact me. We look forward to your response.

Sincerely,

Authors

Review Comments to the Author 

Reviewer #1: The authors have addressed the issues raised by reviewers. There are, however, a few editorial corrections to make. These include:

1) Study setting, second statement (page 6): Rephrase to read: “At the global level, these include …” Also, delete “of 2002” at the end of the UN General Assembly on Special Session on Children, and “at a global level” at the end of the statement since it is unnecessary repetition of the phrase used at the beginning. This has been edited

It now reads, “At the global level these include the 1994 United Nations at the International Conference on Population and Development (ICPD) [26], the 2002 United Nations General Assembly Special Session on Children [27]…”

2) Study setting, third statement (page 6): Rephrase to read: “At the global level …” and delete one “the” after “adopted” to avoid unnecessary repetition. This has been edited

It now reads, “At the regional level, in Africa, Kenya has adopted the Maputo Protocol”

3) Study setting, fourth statement (page 6): Insert “the” before “National Youth Policy” for the statement to read well. This has been edited

It now reads, “Locally the 2010 Constitution of Kenya, the National Youth Policy (2007)…”

4) Ideas: Framing the problem, second quote (pages 14-15): Based on the authors’ response to earlier comment regarding the clarity of the quote, they need to add “[comprehensive sexuality education]” in brackets as shown after “invest more” so that it is clear that the participant was referring to investing in comprehensive sexuality education rather than in programs to prevent pregnancy among adolescents generally. This has been edited

It now reads, “…So I think that both culturally and religiously, there is that feeling that if you invest more [in comprehensive sexuality education] then, they will now know that it is their right (State Corporation; IDI_I0).”

5) Issue characteristics, fourth quote (pages 19-20): Based on the authors’ response to an earlier comment regarding the quote, it should then read: “And the reason for failure ...” This has been edited

It now reads, “I think that is where the challenge lies. And the reason for failure to optimize the services for young people is actually because resources are not there….”

6) Political contexts, first quote (pages 20-21): Based on the authors’ response to the comment on this quote, if the participant used the word “fresh” to mean conducting community forums afresh, then the authors need to insert “[sic]” after the word to show that it was the participant’s mistake (i.e. we cannot use the word “fresh” as a verb). If this was an artifact of transcription, then the authors should change “fresh” to “conduct afresh”.

Thank you for highlighting this. This has been edited from the transcript to read, “but now you have to have fresh community participation [engagement] forums (Ministry of Youth and Gender; IDI_9). The second “have” had been dropped in transcription.

7) Discussion, fourth paragraph, fourth statement (page 27): Change “from 1990 levels by 2015” to “by 2015 from 1990 levels” for the statement to read well. This has been edited

It now reads, “to reduce maternal mortality by three quarters by 2015 from 1990 levels and a set of solutions…”

Reviewer #2: These were my initial comments that were not forwarded to the authors:

"I find this to be an interesting paper and the authors have written it in very clear and easy to read language. Overall, I believe the manuscript, when published will contribute to the understanding of the policy-making and agenda-setting processes. It highlights some of the controversies around adolescence, including the definition of who is an adolescent and the age to include in that definition, the acceptability of providing comprehensive sexuality education to adolescents, the problem of acceptable language, the difficulty of civil society coordination, and the perennial problem of donor-influence in agenda-setting. There are some areas I think the work may require some revisions:

1. State and non-state actor -- It will be useful in the methods section to highlight who the state actors are. I understand that this is in the Table 1, but a high level statement to show the state actors are will be useful rather than expecting the readers to wait until they get to the table or to flip to the table to find the actors and flip back to continue reading.

We have now included the statement: State actors that were targeted included government officials from the ministries of health, youth and gender affairs, devolution and planning, and education. 

2. What do the notations ID1, ID2, ID3... just before a verbatim quotation mean? It lists respondents who had very similar thoughts on the topic. 

First, what is the meaning of "ID"? It means In-depth-interview. We have changed it to IDI throughout the text for clarity.

Second, is the listing various IDs before a verbatim quotation indicative of the relevance of the interviews for a particular participant to the point being made? The listing indicates other respondents who agree with the point being made and had very similar views to the respondent cited. If it is, it will be necessary to state that that is the case. We have included a sentence in the first paragraph under results (page 21) that reads, “The listing of various IDIs before a verbatim quotation correspond to respondents who had similar views to the point being made.” 

 I think the referencing for the quotes can be improved by actually stating the respondent, for example, "Ministry of Youth" or some other form of referencing the quotes. 

We have now included the type of respondents such as Ministry of Youth or International NGO.

3. In a few instances, there are cases of misplaced punctuation marks, which need to be revised. For example before a verbatim quote the authors often place a full-stop and open a bracket to place the ID1, ID2... in without a full-stop at the end of the list. 

We have revised the punctuation to place the full-stop after the closed bracket throughout the paper.

4. It is surprising that the study could find that there are no adolescent SRH champions in Kenya. I wonder if this is simply a matter of the participants that were interviewed. It is really hard to believe that there are no champions. Could the author comment further on this?"

The term policy champion here is used to mean policy entrepreneur as originally defined by John Kingdon. We have changed champion to read entrepreneur throughout the text and provided some context on page 13, which references Kingdon, “Policy entrepreneurs do the process of connecting the problem with a policy solution and the political factors. The entrepreneurs 1) highlight indicators of the problem to dramatize it, 2) Push for one kind of problem definition over another – invite policymakers to see for themselves, and 3) present specific policies as the solution to a problem on the agenda 4) “Soften up” by writing papers, giving testimony, holding hearings or getting press coverage [43]. 

My own thinking regarding this question is in line with what Kingdon says—that entrepreneurs do more than push, push, and push for their proposals or for their conception of problems. They also lie in wait – for a window to open. In the process of leaping at their opportunity, they play a central role in coupling the streams at the window.” This kind of person has not yet emerged in the Kenyan adolescent SRH space. Typically this can be someone like the first lady, a charismatic politician with a personal history or a local or international celebrity. My thoughts are that the potential entrepreneurs e.g. politicians are too risk averse to engage with a controversial issue (page 26). It is easier to champion improved access to maternal services and providing fistula services as the first lady of Kenya has done with the beyond zero campaign but without mentioning the adolescent and “hoping/crossing fingers” that they get addressed

---

## [Editor Report · Decision Letter 2]

27 Nov 2019

Challenges to generating political prioritization for adolescent sexual and reproductive health in Kenya: a qualitative study

PONE-D-19-27067R2

Dear Dr. Onono,

We are pleased to inform you that your manuscript has been judged scientifically suitable for publication and will be formally accepted for publication once it complies with all outstanding technical requirements.

With kind regards,

Joshua Amo-Adjei, Ph.D

Academic Editor

PLOS ONE
---

## [Editor Report · Acceptance letter]

11 Dec 2019

PONE-D-19-27067R2 

Challenges to generating political prioritization for adolescent sexual and reproductive health in Kenya: a qualitative study 

Dear Dr. Onono:

I am pleased to inform you that your manuscript has been deemed suitable for publication in PLOS ONE. Congratulations! Your manuscript is now with our production department. 

With kind regards,

on behalf of

Dr. Joshua Amo-Adjei 

Academic Editor

PLOS ONE